# Broadband microwave detection using electron spins in a hybrid diamond-magnet sensor chip

Joris J. Carmiggelt [1], Iacopo Bertelli [1], Roland W. Mulder[1], Annick Teepe[1], Mehrdad Elyasi[2], Brecht G. Simon [1], Gerrit E. W. Bauer[1,2], Yaroslav M. Blanter[1] & Toeno van der Sar [1] ✉

Quantum sensing has developed into a main branch of quantum science and technology. It aims at measuring physical quantities with high resolution, sensitivity, and dynamic range. Electron spins in diamond are powerful magnetic field sensors, but their sensitivity in the microwave regime is limited to a narrow band around their resonance frequency. Here, we realize broadband microwave detection using spins in diamond interfaced with a thin-film magnet. A pump field locally converts target microwave signals to the sensor-spin frequency via the non-linear spin-wave dynamics of the magnet. Two complementary conversion protocols enable sensing and high-fidelity spin control over a gigahertz bandwidth, allowing characterization of the spin-wave band at multiple gigahertz above the sensor-spin frequency. The pump-tunable, hybrid diamond-magnet sensor chip opens the way for spin-based gigahertz material characterizations at small magnetic bias fields.

Electron spins associated with nitrogen-vacancy (NV) defects in diamond are magnetic field sensors that provide high spatial resolution and sensitivity at room temperature[1,2]. They have been used to study nuclear magnetic resonance at the nanoscale[3,4], bio-[5], paleo-[6], and solid-state magnetism[7], and electric currents in quantum materials[8,9]. Most of these applications focus on detecting magnetic fields in the 0–100 megahertz (MHz) frequency range, in which a toolbox of spin-control techniques enables high sensitivity and a tunable detection frequency without requiring a specific electron spin resonance (ESR) frequency[1]. In contrast, NV-based sensing in the microwave regime [1–100 gigahertz (GHz)] currently relies on tuning the ESR to the frequency of interest using a magnetic bias field[10]. This bias field changes the properties of e.g., magnetic or superconducting samples under study[11,12], for instance by altering their excitation spectrum, which limits its application in materials science. Furthermore, the field must be on the Tesla scale for operation in the 10–100 GHz range[13], making the required magnets large and slow to adjust, precluding the small sensor packaging desired for technological applications.

Here, we enable broadband spin-based microwave sensing by interfacing a diamond chip containing a layer of NV sensor spins with a thin-film magnet. The central concept is that the non-linear dynamics of spin waves–the collective spin excitations of the magnetic film[14]–locally convert a target signal to the NV ESR frequency under the application of a pump field (Fig. 1a, b). We realize a ~1-GHz detection bandwidth at fixed magnetic bias field via four-spin-wave mixing, and microwave detection at multiple GHz above the ESR frequency via difference-frequency generation. The pump-tunable detection frequency enables characterizing the spin-wave band structure despite a multi-GHz detuning and provides insight into the non-linear spin-wave dynamics limiting the conversion process. Furthermore, the converted microwaves are highly coherent, enabling high-fidelity control of the sensor spins via off-resonant drive fields.

## Results

### Sensor platform

Our hybrid diamond-magnet sensor platform consists of an ensemble of near-surface NV spins in a diamond membrane positioned onto a

[1]Department of Quantum Nanoscience, Kavli Institute of Nanoscience, Delft University of Technology, Lorentzweg 1, 2628 CJ Delft, The Netherlands. [2]Advanced Institute for Materials Research (WPI-AIMR), Tohoku University, Sendai 980–8577, Japan. ✉e-mail: T.vanderSar@tudelft.nl

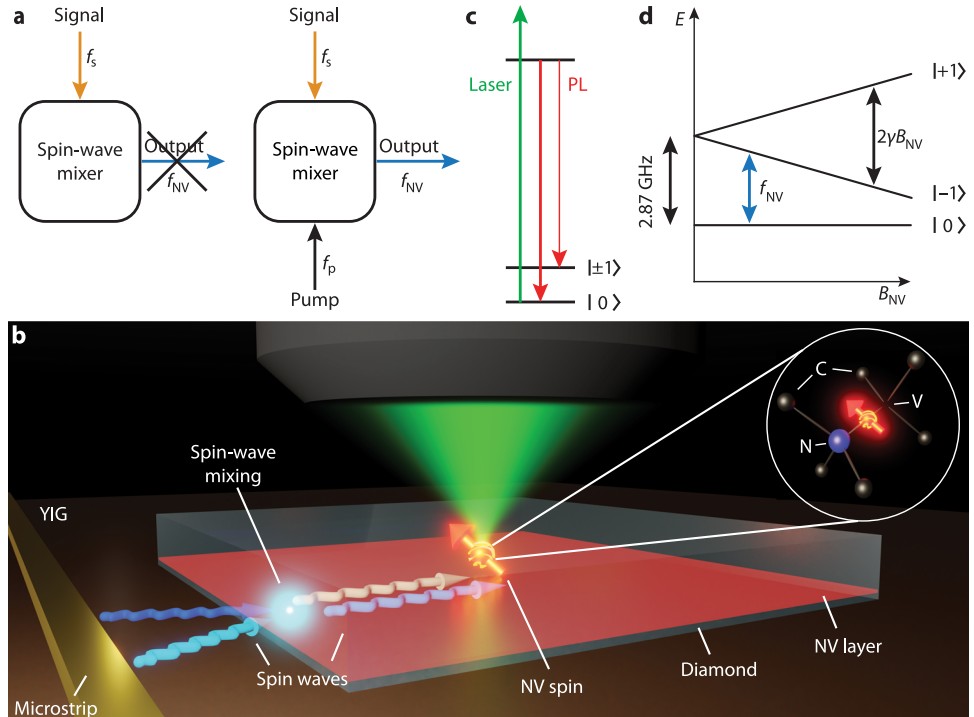

**Fig. 1 | Detecting microwave magnetic fields using spins in diamond via on-chip spin-wave-mediated frequency conversion. a** Idea of the experiment. A 'spin-wave mixer' uses a pump to convert a microwave signal at frequency $f_s$ to an output frequency $f_{NV}$ that is detectable by nitrogen-vacancy (NV) sensor spins in diamond. **b** Sketch of the setup. A diamond with NV centers implanted ~10–20 nm below its surface is placed onto a film of yttrium iron garnet (YIG, thickness: 235 nm). A microstrip delivers the signal and pump microwaves, which excite spin waves in the YIG. Spin-wave mixing enables detection of the signal field by converting its frequency to the NV electron spin resonance (ESR) frequency. Inset: Atomic structure of an NV center in the diamond carbon (C) lattice. **c** Initialization and readout of the NV spins is achieved through excitation by a green laser and detection of the red photoluminescence (PL). The PL is stronger in the $m_s = |0\rangle$ state than in the $m_s = |\pm1\rangle$ states. **d** NV spin energy levels in the electronic ground state. A magnetic field $B_{NV}$ along the NV axis splits the $m_s = |\pm1\rangle$ states via the Zeeman interaction. From the four possible configurations in the diamond lattice, we use the NV orientation with in-plane projection parallel to the stripline. $f_{NV}$ denotes the $|0\rangle \leftrightarrow |-1\rangle$ ESR transition frequency.

thin film of yttrium iron garnet (YIG)—a magnetic insulator with low spin-wave damping[14] (Fig. 1b). A stripline delivers the "two-color" signal and pump microwave fields to the YIG film, in which they excite spin waves at the signal and pump frequencies, $f_s$ and $f_p$, respectively. The frequency-converted microwaves at the ESR frequency $f_{NV}$ are detected by measuring the spin-dependent NV photoluminescence under green laser excitation ("Methods" and Fig. 1c). The ESR frequency is fixed by an external magnetic bias field $B_{NV}$ (Fig. 1d).

## Microwave detection via four-spin-wave mixing

Our first detection protocol harnesses degenerate four-spin-wave mixing[15–20]—the magnetic analog of optical four-wave mixing (Fig. 2a). In the quasiparticle picture, this process corresponds to the scattering of two "pump" magnons into a "signal" magnon and an "idler" magnon at frequency $f_i = 2f_p - f_s$. This conversion enables the detection of a microwave signal that is detuned from the ESR frequency, which would be otherwise invisible in the optical response of the NV centers (Fig. 2b). By tuning the frequency of the pump, we enable the detection of signals of specific microwave frequencies (Fig. 2c).

We characterize the bandwidth of the four-wave-mixing detection scheme by measuring the NV photoluminescence contrast as a function of the microwave signal frequency and magnetic bias field. As in Fig. 2b, when the pump field is switched off, we only detect signals resonant with $f_{NV}$ (Fig. 2d). In contrast, when the pump is switched on, a broad band of frequencies becomes detectable (Fig. 2e). The bandwidth $\Delta f$ of ~1 GHz is limited from below by the ferromagnetic resonance (FMR), the spatially homogenous spin-wave mode below which spin waves cannot be excited in our measurement geometry, and from above by the limited efficiency of our 5-micron-wide stripline to excite

high-momentum spin waves. As such, the bandwidth can be extended by using narrower striplines or magnetic coplanar waveguides[21].

At 14 dBm signal and pump power, consecutive mixing processes generate higher-order idler modes at discrete and equally spaced frequencies (Fig. 2f). Motivated by the success of their optical counterparts in high-precision spectrometry[22], such "spin-wave frequency combs" are of great interest because of potential applications in microwave metrology[20,23,24]. We use the spin-wave comb to realize sensitivity to multiple microwave frequencies by detecting the $n$-th order idler frequency,

$$f_i^{(n)} = (n+1)f_p - nf_s \tag{1}$$

when it is resonant with the ESR frequency (Fig. 2f, upper inset). An increasing number of idler modes appears with increasing drive power (Supplementary Fig. 3), such that at large powers we resolve up to the $n = 10$th idler order (Fig. 2f, bottom inset). The shift of the idler frequency is amplified by the integer $n$ over the shift of the signal frequency (Eq. 1), leading to a $1/n$ decrease in the linewidth of the NV ESR response[24] (Fig. 2f) and a correspondingly enhanced ability to resolve closely spaced signal frequencies.

## Coherent spin control via four-spin-wave mixing

In addition to enabling off-resonant quantum sensing, the idlers also provide a resource for off-resonant control of spin- or other quantum systems. The resolving of the NV's 3-MHz hyperfine splitting in the idler-driven ESR spectrum (Fig. 3a) evidences the high coherence of the microwave field emitted by the idler spin wave, implying that the linewidth is determined by the drive rather than the spin-wave

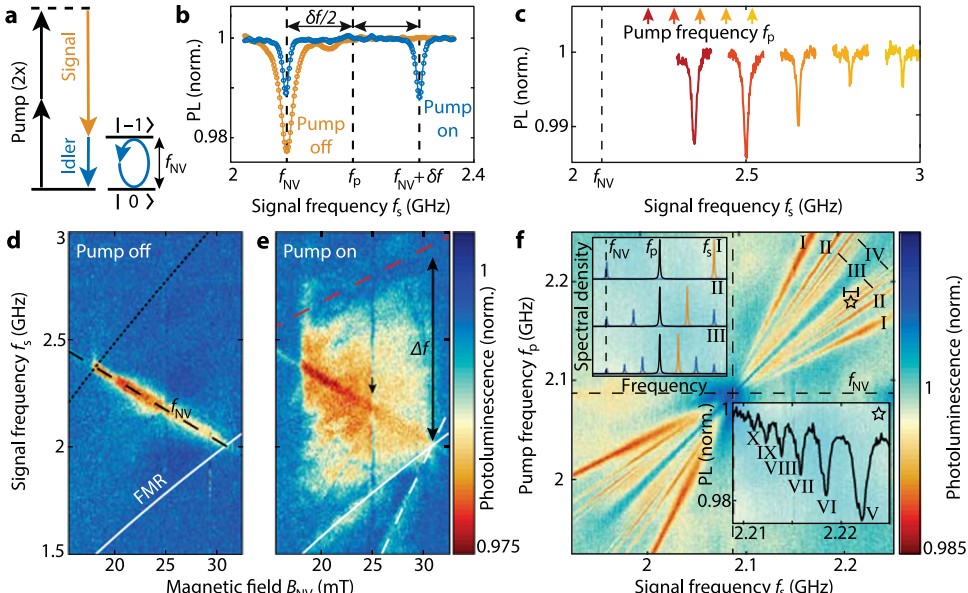

**Fig. 2 | Microwave detection via four-spin-wave mixing and frequency combs.**
**a** Energy diagram of four-spin-wave mixing. The signal at frequency $f_s$ stimulates the conversion of the pump to the idler at $f_{NV}$. **b** Normalized NV photoluminescence (PL) versus $f_s$. Without pump (orange data), an ESR dip is only observed when $f_s = f_{NV}$. With pump at $f_p = f_{NV} + \delta f/2$ (blue data), a signal at $f_s = f_{NV} + \delta f$ becomes detectable. **c** Tuning the pump (colored arrows) shifts the detectable signal frequency, observed through the shifting ESR dips (matching colors). **d** Normalized NV PL vs $f_s$ and magnetic field in the absence of a pump. Only signals at $f_{NV}$ (dashed black line) can be detected. Dotted black line: Frequency above which three-magnon scattering limits the spin-wave amplitude[33]. White line:

Ferromagnetic resonance (FMR) frequency (Supplementary Note 1). **e** Applying a pump at $f_p = (f_s + f_{NV})/2$ opens a detection window from the FMR up to the second node (dashed red line) in the Fourier spectrum of the stripline field (Supplementary Fig. 1). (Dashed) white line: Signal (Pump) drives FMR. Black arrow: Line of reduced contrast caused by scattering into the first perpendicular standing spin-wave mode[11]. **f** Spin-wave comb observed in the PL versus $f_s$ and $f_p$. Data is normalized (Supplementary Fig. 2). Upper inset: Spectrum (sketch) illustrating the detection of idlers I–III (black: pump, orange: signal, blue: idlers). Lower inset: Linecut along the small black line at the star in the main panel, showing idlers up to the tenth order.

damping[24]. This allows driving coherent NV spin rotations (Rabi oscillations) by pulsing the pump with varying duration $\tau$ (Fig. 3b).

Remarkably, these Rabi oscillations respond to externally applied microwaves that are detuned by hundreds of MHz from the ESR frequency (Fig. 3c). Such magnon-mediated, off-resonant Rabi control is a new instrument in the toolbox of spin-manipulation techniques, providing universal off-resonant quantum control with potential applications in quantum information processing. The idler-driven Rabi frequency exceeds the signal-induced AC Stark shift[25] by about an order of magnitude for the same off-resonant signal power (Supplementary Fig. 4). The decrease of the Rabi frequency with increasing detuning $\delta f$ (Fig. 3c) is the combined result of a reduced spin-wave excitation efficiency at higher frequency, because the stripline is less efficient in exciting spin waves with short wavelengths (Supplementary Note 2), and a reduced spin-wave scattering strength due to the increasing momentum mismatch between signal and pump spin waves[17–19].

Since the Rabi frequency depends linearly on the idler amplitude[11], it provides insight into the magnetization dynamics in the film. As expected, the idler amplitude initially grows with increasing signal and pump power[15,20], but then reaches a maximum and starts to decrease (Fig. 3d). We attribute the decrease to Suhl instabilities of the second type[16]: Both signal and pump modes decay into a pair of high-momentum magnons beyond a certain threshold amplitude, which drains energy from the idler mode. This interpretation is supported by a model of the four-wave interactions between the dominant two idler modes, the signal and pump modes, and the two pairs of high-momentum "Suhl" magnons (Supplementary Figs. 5 and 6). The intermode coupling is induced by exchange and dipolar interactions, as well as crystalline anisotropy, and follows from the leading-order terms in the Holstein-Primakoff expansion[17]. Based on the

interacting eight-mode Hamiltonian we compute the steady-state dynamics of the idler mode as a function of pump and signal power (Fig. 3e, Supplementary Note 4), which qualitatively reproduces the observed power dependence in Fig. 3d.

## Microwave detection via difference-frequency generation

Our second detection protocol relies on difference-frequency generation, which enables down-conversion of GHz signals to MHz frequencies accessible to established quantum sensing techniques[1]. The difference frequency is generated by the longitudinal component of the magnetization under the driving of two spin waves of different frequencies[26] (Fig. 4a, Supplementary Note 5). In contrast to the four-wave mixing protocol, the converted frequency does not have to lie within the spin-wave band. By tuning the ESR frequency into resonance with the difference frequency (Fig. 4b), we detect microwave signals that are detuned by several gigahertz when $f_p - f_s = \pm f_{NV}$ (Fig. 4c). Alternatively, AC magnetometry protocols can provide difference-frequency detection with enhanced sensitivity at arbitrary bias fields[1]. We only observe ESR contrast when both $f_s$ and $f_p$ are above the FMR (Fig. 4d), confirming that the conversion is mediated by spin waves in the YIG. We anticipate the conversion process can also be applied in other magnetic materials to characterize high-frequency magnetic band structures that would otherwise be out of reach for NV magnetometry (Supplementary Note 6). Similar to Fig. 2e, the conversion is limited by the spin-wave excitation efficiency, which explains the observation of the largest ESR contrast for long-wavelength spin waves (i.e., just above the FMR).

## Discussion

We demonstrated magnon-mediated, spin-based sensing of microwave magnetic fields over a gigahertz bandwidth at fixed magnetic bias field. The frequency of the pump determines the detection frequency,

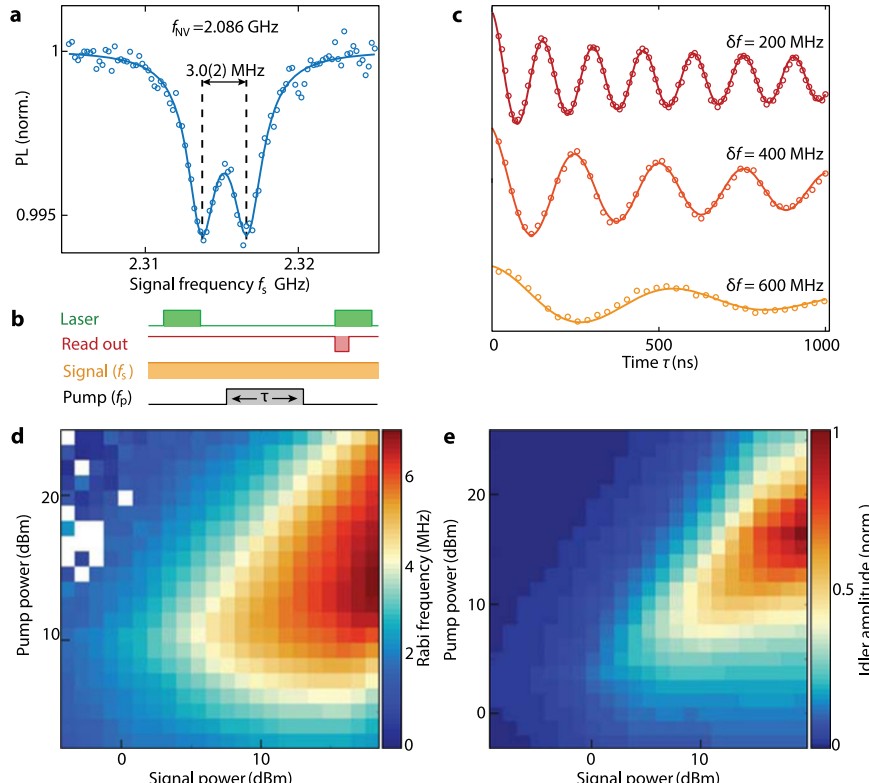

**Fig. 3 | Off-resonant quantum control of NV spins via frequency conversion based on four-spin-wave mixing. a** Idler-driven NV ESR spectrum with the ESR frequency at $f_{NV} = 2.086$ GHz and the pump at $f_P = 2.2$ GHz. The narrow linewidth of the idler allows resolving the 3-MHz hyperfine splitting associated with the $^{15}$N nucleus. **b** Pulse sequence for driving coherent NV spin rotations (Rabi oscillations) with an off-resonant signal field. The pulsed pump and continuous-wave signal generate a pulsed idler at $f_{NV}$ that drives Rabi oscillations. **c** Optically-detected Rabi oscillations driven by the first-order idler mode for different detuning $\delta f = f_S - f_{NV}$. **d** Frequency of the idler-driven Rabi oscillations versus power of the pump ($f_P = 2.2$ GHz) and signal ($f_S = 2.314$ GHz). The Rabi frequency initially increases with both signal and pump power, but then decreases because of spin-wave instabilities. This non-monotonic behavior is reproduced by numerical calculations of the normalized idler amplitude in (**e**) (details in Supplementary Note 4).

with a detection range that is limited only by the frequencies at which spin waves can be excited efficiently. The coherent nature of the frequency conversion enables coherent manipulation of solid-state spins via off-resonant drive fields, as demonstrated here for spins in diamond. This coherence allows combining with advanced spin-manipulation protocols such as heterodyne or dressed-state sensing[27–29] to further enhance the detection capabilities, and opens the way for applications in hybrid quantum technologies[30]. Wide-field readout of NV centers in a larger sensing volume would enhance the microwave sensitivity, which is ultimately limited by thermal spin-wave noise. We envision the detection of free-space microwaves using on-chip microwave-to-spin-wave transducers[31] such as stripline resonators, and the characterization of local microwave generators such as spin-torque oscillators by combining with a suitable magnetic material[32] and applying a pump field. Imaging of the spatial magnetization dynamics generated by spin-wave mixing using scanning-NV magnetometry could provide insight into the spin-wave dispersion and interactions with nanoscale sensitivity[2]. The demonstrated hybrid diamond-magnet sensor platform enables broadband microwave characterization without requiring large magnetic bias fields and opens the way for probing high-frequency magnetic spectra of new materials, such as van-der-Waals magnets.

## Methods
### Experimental setup
The NV photoluminescence is read out using a confocal microscope described in ref. [11]. The NV-YIG chip and its fabrication were described in ref. [33]. It consists of a $2 \times 2 \times 0.05$-mm$^3$ diamond membrane

with an estimated near-surface NV density of $10^3/\mu m^2$ placed on top of a 235-nm-thick YIG film grown using liquid phase epitaxy on a 500-μm-thick GGG substrate (Matesy GmbH). The diamond-YIG separation distance is ~2 μm, limited by small particles (such as dust) between the diamond and the YIG surfaces. The signal and pump microwaves are generated by two Rohde & Schwarz microwave sources (SGS100A), combined by a Mini-Circuits power combiner (ZFRSC-123-S+, total loss: ~ −10 dB) and amplified by an AR amplifier (30S1G6, amplification: ~44 dB). All measurements were performed at room temperature.

### NV microwave magnetometry
The four NV-center families are sensitive to microwave magnetic fields at their electron spin resonance (ESR) frequencies, which are determined by the magnetic bias field $\mathbf{B}_{NV}$ via the NV spin Hamiltonian $H = DS_z^2 + \gamma \mathbf{B}_{NV} \cdot \mathbf{S}$, with $D = 2.87$ GHz the zero-field splitting, $\gamma = 28$ GHz/T the electron gyromagnetic ratio and $S_{i \in \{x,y,z\}}$ the $i$th spin-1 Pauli matrix. In this work, we align the field along one of the NV orientations, such that this "on-axis" family has $|0\rangle \leftrightarrow |\pm 1\rangle$ ESR frequencies given by $D \pm \gamma B_{NV}$ (with $B_{NV} = |\mathbf{B}_{NV}|$). For the other three "off-axis" families, the bias field is equally misaligned by ~71° due to crystal symmetry, leading to the ESR frequency plotted in Fig. 4b (labeled "Off-axis"). The photoluminescence dips were recorded using continuous-wave microwaves and non-resonant optical excitation at 515 nm. For the Rabi oscillations, we first initialize the NV spin in the $|0\rangle$-state via a ~1-μs green laser pulse, then we drive the spin using an idler pulse and finally we read out the NV photons in the first 300–400 ns of a second laser pulse.

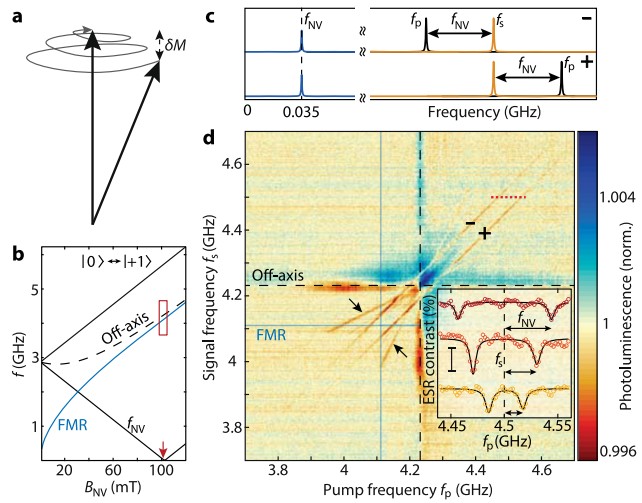

**Fig. 4 | Microwave detection via difference-frequency generation. a** Sketch of the spiraling precession of the magnetization $M$ when driven by microwave frequencies $f_s$ and $f_p$. The longitudinal component of the magnetization oscillates with amplitude $\delta M$ at the difference frequency ($|f_s - f_p|$), which is detected when resonant with $f_{NV}$. **b** Field dependence of the FMR (solid blue line) and the NV ESR frequencies, with the solid (dashed) black line corresponding to the on-axis (off-axis) NV families. $f_{NV}$ of the on-axis family enters the MHz regime near $B_{NV} = 100$ mT (red arrow). **c** Sketch of the frequency spectrum. Difference-frequency generation creates a detectable signal at $f_{NV}$ when the pump frequency is at $f_p = f_s \pm f_{NV}$. **d** Photoluminescence versus $f_s$ and $f_p$ at $B_{NV} = 101.3$ mT, such that $f_{NV} = 35$ MHz [red box in (**b**)]. The parallel, diagonal lines [labeled + and − as in (**c**)] indicate the difference-frequency detection. Also visible are idler spin waves (indicated by the arrows) generated by four-wave mixing at the off-axis ESR frequency (dashed black lines). The data is normalized, leaving artefacts at the off-axis frequency ("Methods"). Inset: Line traces at $f_s = 4.5$ GHz (dotted red line in main panel) for different $B_{NV}$, showing the shift of the frequency difference resonant with $f_{NV}$. Scalebar: 0.1% ESR contrast.

## Data processing

The data presented in Figs. 2f and 4d are normalized by the median of each row and column (Supplementary Fig. 2).

## Data availability

The numerical data plotted in the figures in this work are available at Zenodo with identifier https://doi.org/10.5281/zenodo.6543615.

## Code availability

The codes used for the numerical calculation of the idler amplitude are available upon request.

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

## Acknowledgements

We thank M. N. Ali for commenting on the manuscript. This work was supported by the Dutch Research Council (NWO) through the Frontiers of Nanoscience (NanoFront) program (J.J.C., I.B., and T.v.d.S.), the NWO Projectruimte grant 680.91.115 (B.S. and T.v.d.S.), the Kavli Institute of Nanoscience Delft (T.v.d.S. and Y.M.B.), and the Japan Society for the Promotion of Science (JSPS) by Kakenhi Grant # 19H00645 (G.B.).

## Author contributions

J.J.C. and T.v.d.S. conceived the experiment. I.B. built the setup and fabricated the sample. R.W.M., J.J.C., I.B., and A.T. performed the measurements and analyzed the data. M.E., Y.M.B., and G.E.W.B. developed the theoretical model for the idler amplitude, B.G.S. fabricated the diamond membrane, J.J.C., T.v.d.S., I.B., and A.T. wrote the manuscript with help from all coauthors.

## Competing interests

The authors declare no competing interests.
