## [Peer Review File · Nature Communications]

Broadband microwave detection using electron spins in a hybrid diamond-magnet sensor chipEditorial Note: This manuscript has been previously reviewed at another journal that is not operating a transparent peer review scheme. This document only contains reviewer comments and rebuttal letters for versions considered at *Nature Communications*.

REVIEWER COMMENTS

Reviewer #1 (Remarks to the Author):

Overall, I stand by my previous generally positive review of the manuscript. The authors have indeed improved the manuscript greatly on most points that I have that were raised.

Only regarding my concern with respect to the supposed usefulness of the technique for metallic ferromagnets at frequencies of up to 100 GHz I am still not happy. I am well aware of all the arguments made by the authors on this point in the rebuttal. I am afraid I am still not convinced. Therefore, I would like to see this discussion modified. In my view Fig.4.E is misleading. In this manuscript it has not at all been shown that this will work with these materials (only low GHz frequencies with YIG are addressed). YIG is really a very special case and only YIG was used in the experiments. Therefore, if no further experiments with the metallic ferromagnets are presented I suggest to move this very speculative discussion to the supplementary material and mention the potential application only briefly in the manuscript.

Besides this main concern I fully support the publication of the manuscript in Nature Communications.

Reviewer #3 (Remarks to the Author):

The authors have carefully and satisfactorily addressed all of the comments by the referees. In my opinion, the paper should be published as is.

Reply to referee 1

Reviewer #1 (Remarks to the Author):

Overall, I stand by my previous generally positive review of the manuscript. The authors have indeed improved the manuscript greatly on most points that I have that were raised.

Only regarding my concern with respect to the supposed usefulness of the technique for metallic ferromagnets at frequencies of up to 100 GHz I am still not happy. I am well aware of all the arguments made by the authors on this point in the rebuttal. I am afraid I am still not convinced. Therefore, I would like to see this discussion modified. In my view Fig.4.E is misleading. In this manuscript it has not at all been shown that this will work with these materials (only low GHz frequencies with YIG are addressed). YIG is really a very special case and only YIG was used in the experiments. Therefore, if no further experiments with the metallic ferromagnets are presented I suggest to move this very speculative discussion to the supplementary material and mention the potential application only briefly in the manuscript.

Besides this main concern I fully support the publication of the manuscript in Nature Communications.

We thank the referee for the positive review. Following the referee's request, we have moved Fig. 4E and the associated discussion of metallic ferromagnets to the supplement. In addition, we changed the last sentence of the abstract. Specifically, we removed:

"This range could be extended to the 10-100 GHz scale using materials with a larger magnetization that increases the spin-wave group velocity or crystal anisotropies that increase the spin-wave gap (Fig. 4E). The four-spin-wave mixing process, spin-wave frequency combs and oscillations of the longitudinal magnetization used in our protocols have already been observed in other magnets, such as permalloy (Py), Fe, and CoFe18–20,27. The increased spin-wave damping in these materials compared to YIG reduces the spin-wave amplitudes, but this is partially compensated by a larger saturation magnetization that increases the stray fields."

from the Discussion section of the main text. In addition, we changed the final sentence of the abstract from:

"The pump-tunable, hybrid diamond-magnet sensor chip opens the way for spin-based sensing in the 100-gigahertz regime at small magnetic bias fields."

to

"The pump-tunable, hybrid diamond-magnet sensor chip opens the way for spin-based gigahertz material characterizations at small magnetic bias fields."

Reply to referee 3

Reviewer #3 (Remarks to the Author):

The authors have carefully and satisfactorily addressed all of the comments by the referees. In my opinion, the paper should be published as is.

We thank the referee for recommending publication as is.